# Discovery of Novel Thiazolidinedione-Derivatives with Multi-Modal Antidiabetic Activities In Vitro and In Silico

**DOI:** 10.3390/ijms24033024

**Published:** 2023-02-03

**Authors:** Charles Arineitwe, Ogunyemi Oderinlo, Matshawandile Tukulula, Setshaba Khanye, Andile Khathi, Ntethelelo Sibiya

**Affiliations:** 1Pharmacology Division, Rhodes University, Grahamstown 6140, South Africa; 2Department of Chemistry, Faculty of Science, Federal University, Otuoke Bayelsa 562103, Nigeria; 3School of Chemistry and Physics, College of Agriculture, Engineering Sciences, University of KwaZulu-Natal, Durban 4000, South Africa; 4Pharmaceutical Chemistry Division, Faculty of Pharmacy, Rhodes University, Makhanda 6140, South Africa; 5School of Laboratory Medicine and Medical Sciences, College of Health Sciences, University of KwaZulu-Natal, Durban 4000, South Africa

**Keywords:** synthesis, pharmacological screening, thiazolidinedione-derivatives, diabetes mellitus, therapeutic targets

## Abstract

Diabetes mellitus (DM) and related complications continue to exert a significant burden on health care systems globally. Although conventional pharmacological therapies are beneficial in the management of this metabolic condition, it is still necessary to seek novel potential molecules for its management. On this basis, we have synthesised and evaluated the anti-diabetic properties of four novel thiazolidinedione (TZD)-derivatives. The TZD derivatives were synthesised through the pharmacophore hybridisation strategy based on *N*-arylpyrrole and TZD. The resultant derivatives at different concentrations were screened against key enzymes of glucose metabolism and glucose utilisation in the liver (HEP-G2) cell line. Additionally, peroxisome proliferator-activated receptor-γ activation was performed through docking studies. Docking of these molecules against PPAR-γ predicted strong binding, similar to that of rosiglitazone. Hence, TZDD2 was able to increase glucose uptake in the liver cells as compared to the control. The enzymatic inhibition assays showed a relative inhibition activity; with all four derivatives exhibiting ≥ 50% inhibition activity in the α-amylase inhibition assay and a concentration dependent activity in the α-glucosidase inhibition assay. All four derivatives exhibited ≥30% inhibition in the aldose reductase inhibition assay, except TZDD1 at 10 µg/mL. Interestingly, TZDD3 showed a decreasing inhibition activity. In the dipeptidyl peptidase–4 inhibition assay, TZDD2 and TZDD4 exhibited ≥20% inhibition activity.

## 1. Introduction

Diabetes mellitus prevalence continues to show an upward trajectory, exerting a serious cost burden on healthcare systems all over the world [1,2,3,4,5]. Current treatments used in the clinical care and management of diabetes mellitus (DM) aim to obtain and maintain blood glucose concentrations as close to normal as possible to avoid wide glycemic fluctuations. Furthermore, the goal of the treatment is to prevent the development of long-term complications. In type 2 diabetes mellitus, current treatments include insulin, amylin and incretin mimetics, dipeptidyl peptidase-4 inhibitors, glinides, sulphonylureas, biguanides, thiazolidinediones, alpha-glucosidase inhibitors, and sodium-dependent glucose co-transporter-2 inhibitors.

Type 2 diabetes treatment options are increasingly expensive, not readily available, and frequently have numerous side-effects such as hypoglycemia, diarrhoea, weight gain, and abnormal liver function [5,6]. Hence, new generations of small molecules are being investigated with an aim to improve efficacy and safety profiles. Among these are thiazolidinedione (TZD) derivatives. According to current literature, Iqbal et al. synthesised novel TZD derivatives that showed insulin sensitising properties [7]. Also, Bhattarai et al. synthesised benzylidene-2,4-thiazolidinedione derivatives that were equally potent as PTP1B [8]. Maccari et al. also reported new aldose reductase inhibitors (ARIs) through in vitro evaluation of a series of 5-arylidene-3-(3,3,3-trifluoro-2-oxopropyl)-2,4-thiazolidinediones at low-micromolar doses [9].

These TZD derivatives are potential antidiabetic agents that have exhibited various modes of action such as PPAR-γ activation, aldose reductase inhibition, and protein tyrosine phosphate-1B inhibition [10]. In this study, we synthesised TZD derivatives by exploring the use of *N*-heterocyclic amines as an appendant on the *N*-3 of the TZD scaffold since several compounds containing these structural scaffolds are known to exert biological activity. Earlier studies have shown that TZD-derived hybrids with an *N*-functionalized side chain on the *N*-3 nitrogen TZD ring display improved activity, and such modifications exhibit low toxicity towards primary cultured non-malignant human hepatocytes [11,12]. Of particular interest was the report that the incorporation of *N*-heterocyclic structures such as pyrrolidinyl, piperidinyl, morpholinyl, and thiomorpholinyl groups into a molecule resulted in compounds with enhanced metabolic stability and increased activity [13].

Based on the above literature, the new TZD derivatives were synthesised and evaluated on diabetic therapeutic targets including α-amylase, α-glucosidase, aldose reductase, PPAR-γ, protein tyrosine phosphatase-1B, and DPP4. Lastly, the effect of these derivatives on some key glucose-handling tissues was investigated.

## 2. Results

### 2.1. α-Amylase Activity

Figure 1 shows the inhibitory properties of TZD derivatives (10, 20, 30, 40, and 50 µg/mL) on α-amylase activity. TZDD3 showed concentration-dependent inhibitory activity, with 30 µg/mL and above showing statistical significance (*p* < 0.05) by comparison to the control. Similarly, TZDD1, 2, and 4 showed inhibitory activity, with all the concentrations showing significance (*p* < 0.05) by comparison to the control. A significant (*p* < 0.05) inhibitory activity was observed with acarbose in all concentrations. Of the four TZD derivatives investigated, TZDD2 showed better inhibitory activity in α-amylase inhibition assay with an IC_50_ value of 18.24 µg/mL (Table 1).

### 2.2. α-Glucosidase Activity

Figure 2 shows the inhibitory properties of TZD derivatives (10, 20, 30, 40, and 50 µg/mL) on α-glucosidase activity. TZDD1-4 showed inhibitory activity, with all concentrations showing statistical significance (*p* < 0.05) by comparison to the control. A significant (*p* < 0.05) inhibitory activity was observed with acarbose in all concentrations. Out of the four TZD derivatives investigated, TZDD3 had a more potent inhibitory activity in the α-glucosidase inhibition assay, as evidenced by a smaller IC_50_ (Table 1). The kinetic analysis of α-glucosidase inhibition, using Michaelis-Menten and the Lineweaver-burk plot kinetic analysis (Figure 3 and Figure 4) by TZDD3, demonstrated a decrease in *Vmax* and *Km* in comparison with the uninhibited reaction (Table 2).

### 2.3. Aldose Reductase Activity

Figure 5 shows the effect of TZD derivatives (10, 20, 30, 40, and 50 µg/mL) on the inhibition of aldose reductase activity. All the TZD derivatives showed inhibitory activity, with TZDD1 showing a concentration-dependent inhibitory activity. TZDD1-4 showed inhibitory activity, with all concentrations showing significance (*p* < 0.05) by comparison to the control. As anticipated, a significant (*p* < 0.05) inhibitory activity was observed with quercetin in all concentrations. Of the four TZD derivatives investigated, TZDD1 showed better inhibitory activity in this aldose reductase inhibition assay with an IC_50_ value of 27.54 µg/mL (Table 1).

### 2.4. Protein Tyrosine Phosphatase 1B Activity

Figure 6 shows the effect of TZD derivatives (10, 20, 30, 40, and 50 µg/mL) on inhibition of protein tyrosine phosphatase 1B activity. TZDD1, 2, and 4 showed weak inhibitory activity, with TZDD4 showing inhibitory activity, with all concentrations showing significance (*p* < 0.05) by comparison to the control. Of the four TZD derivatives investigated, TZDD2 was more potent in the inhibition of protein tyrosine phosphatase with an IC_50_ value of 136.80 µg/mL (Table 1).

### 2.5. DPP-4 Activity

Figure 7 shows the effect of TZD derivatives (10, 20, 30, 40, and 50 µg/mL) on DPP-4 inhibition activity. TZDD2 showed a relatively higher activity, showing significance by comparison to the control in all the concentrations. TZDD1 and 3 showed very weak inhibitory activity with no significance (*p* < 0.05) by comparison to the control. A significant (*p* < 0.05). DPP-4 inhibitory activity was observed with sitagliptin in all concentrations. Of the four TZD derivatives investigated, TZDD2 was more potent in inhibiting DPP-4 activity as evidenced by a smaller IC50 (Table 1). No dose-dependent activity was observed in all four TZDD investigations.

### 2.6. Cell Viability

Figure 8 shows the effect of TZDD 1, 2, 3, and 4 at different concentrations (25, 50, and 100 µg/mL) after 24 h of exposure. By comparison with the control, all the investigated compounds did not affect the cell viability of the HEP-2 cells.

### 2.7. Glucose Uptake

Figure 9 shows the effect of TZDDs (25, 50, and 100 µg/mL) on glucose uptake in liver cells (HEP-G2) at 24 h. TZDD3 showed a significant glucose uptake in comparison to the control at 25 and 50 µg/mL. Other TZDDs demonstrated a very poor glucose uptake effect. As expected, insulin enhanced glucose uptake in the liver cells.

### 2.8. In Silico Determination of the PPAR-γ Activation

PPAR-γ activation by TZDDs and rosiglitazone was demonstrated virtually through the binding energies, RMSD values, and interactions. Figure 10 shows the interaction between the TZD derivatives and the PPAR-γ protein. These derivatives formed various interactions with the PPAR-γ protein residues, including hydrogen bond interactions. In addition, rosiglitazone formed a Ꙥ-cation interaction. Interestingly, the derivatives also showed lower RMSD values than rosiglitazone (Table 3). TZDD4 shows the most similar interaction to the standard (rosiglitazone) used, as seen with its lowest binding energy, estimated inhibition constant, and conformation (Figure 10 and Table 3). TZZD1 and the active site of the protein could not form the two-dimensional conformation, hence they have not been included in the diagram. However, the LBE, RMSD, and EIK have been obtained from docking studies and are indicated in Table 3.

## 3. Discussion

Despite preventative and treatment strategies, the prevalence of diabetes mellitus is inevitably increasing. Therefore, it is still paramount to discover novel pharmacological agents that might exert heightened therapeutic value. In this study, we developed TZD derivatives by exploring the use of *N*-heterocyclic amines as an appendant on the *N*-3 of the TZD scaffold, since several compounds containing these structural scaffolds are known to exert biological activity. Our strategy was to create pharmacophoric units that could target multiple targets for type 2 DM management.

The effect of TZD derivatives on the activities of α-amylase (≥50% inhibition) and α-glucosidase (15–50% inhibition) may suggest that clinically, low doses are required to delay glucose absorption in the gut. The high enzyme affinity (Km = 4.556) (Table 1) of TZDD3, backed by the Lineweaver-Burk plot, suggests a competitive mode of inhibition as achieved by acarbose. This could imply that the active functional group in the derivative competes with the substrate for binding to the active site of the enzyme, thus preventing the breaking down of complex carbohydrates. Several studies have identified hydrogen bonds and hydrophobic interactions with the amino acid residues of α-amylase as the mode of inhibition of compounds that have shown activity against α-amylase [14]. The presence of lipophilic amino acid residues Leu162, Leu165, and Ile235, in the active site of α-amylase could be an essential information regarding hydrophobic interactions with the inhibitor compounds [15]. Furthermore, the existence of functional groups such as the methoxy group, the carboxylic acid group, and the aliphatic CH groups may be crucial for binding [15]. We can also suggest that the lipophilic tail and the methyl groups present in the derivatives form hydrophobic interactions with the amino acid residues of the binding site in the α-amylase, through ԯ-stacking. The ketone group preserved on 4C may also form hydrogen bonds with the amino acid residues, thus making the enzyme-inhibitor complex stronger.

The inhibition activity of the α-glucosidase could be that the derivatives, especially TZDD3, are interacting with the orthosteric binding site (OBS). Acarbose, a standard competitive inhibitor of α-glucosidase, is surrounded by residues of His111, Asp214, Glu276, Asp343, and His348 in the OBS of α-glucosidase [16,17]. Chen et al. demonstrated that Glu426 and Lys155 are particularly crucial for the formation of key hydrogen bonds, and that there were π-π stack interaction between the aromatic ring A of cyanidin and the residue Phe311 of α-glucosidase [18]. Likewise, we can hypothesize that the investigated TZD derivatives are creating interactions with the α-glucosidase residues in the binding site through hydrogen bonds, π-π stack interaction between the aromatic tail rings, and aliphatic chains on the side of the ring with the Phe311 residue.

Arulmozhiraja et al. demonstrated that DPP-4 inhibitors reside inside the hydrophobic cavity consisting of Arg125, Glu205, Glu206, Tyr547, Tyr662, Tyr666, Ser630, and Phe357 [19]. DPP-4 inhibitors interact strongly with the Glu206 and Glu205 amino acids in the S_2_ subsite by forming salt bridges with them. We can therefore suggest that the TZD derivatives did not form strong interaction bonds (weak inhibition observed, Figure 10), either due to their chain length or size, and hence the weak inhibitory effect. DPP4 inhibition has been found to have a beneficial effect extending beyond glycemic control; therefore, novel molecules showing DPP-4 inhibitory properties should be welcomed and investigated further.

It is important to observe that the investigated four TZD derivatives retained TZD moiety, as it has been proven to elicit anti-hyperglycaemic activity through peroxisome proliferator-activated receptor gamma (PPAR-γ) activation [20]. Generally, TZDs form hydrogen bonds with His323 (H4), His449 (H11), and Tyr473 residues of helix 12 of PPAR-γ LBD. The hydrophobic tail moiety of rosiglitazone may also interact with helix 3, 5, 6, and 7 and the β strand, occupying Arm II and Arm III of the LBD, through van der Waals and hydrophobic interactions, which account for the efficiency of binding and potency of the molecule. The central phenyl ring is accommodated beneath helix 3 by hydrophobic interactions [21]. In this study, however, the interactions between rosiglitazone, and the TZD derivatives with PPAR-γ suggest that these derivatives were interacting with another minor binding site. Literature proposes that there are about 32 amino acid residues that can easily form interactions with ligands [22].

With understanding the mode of action and the analysis of binding activity of the investigated TZD derivatives, we may then hypothesise that binding to the PPAR-γ would result in its activation. Indeed, TZDD2 managed to increase glucose uptake in the liver cell lines, which could be explained by the activation of PPAR-γ. However, we also don’t rule out other possible mechanisms, such as the inhibition of PTP-1B, which was also observed in this study, although the inhibition was modest.

Often, the development of diabetes complications is inevitable, despite glycemic control. For this reason, several pathways are targeted in the pathogenesis and progression of diabetes complications such as retinopathy, neuropathy, and nephropathy [23]. In this study, TZDDs exhibited significant inhibitory activity against aldose reductase. Inhibition of this enzyme by the derivatives may therefore be beneficial in the prevention of cellular oxidative stress and redox imbalance, thereby delaying or ameliorating diabetic complications. Quercetin has been demonstrated to possess strong inhibition against enzymes in the polyol pathway and is therefore used widely as a positive control [24]. More importantly, quercetin possesses structural features such as the hydroxyl group at the 3′ position in the B ring, the 2′,3 double bond, and the 4-oxo group in the C ring, which are essential for binding to the active site of the enzymes. Quercetin also inhibits aldose reductase through a variety of mechanisms, including competitive, non-competitive, and uncompetitive inhibition [25,26]. Similarly, the investigated TZD derivatives possess a double bond on the substituent on C5 of the TZD acid head, one oxo group in TZDD1 and 2, and two oxo groups in TZDD3. This can be used to predict that the derivatives will form the required interactions in the binding site, hence the potent inhibitory effect of the TZD derivatives on aldose reductase depicted. Therefore, it can be postulated that the derivatives investigated in this study displayed significant inhibitory properties against aldose reductase due to the mentioned structural features and will need confirmation in an in vivo model of the study of diabetic complications. Taken together, our compounds show moderate activity in several targets for diabetes management. These observations may suggest that TZDDs may exert antidiabetic activity by inhibiting or activating various key targets in DM management, which could be of paramount therapeutic value. For these reasons, further investigations, including in vivo studies, are critical moving forward, to gain holistic therapeutic insights.

## 4. Materials and Methods

### 4.1. Synthesis of Compounds

The synthetic route of the target thiazolidine-2,4-dione carboxamides is illustrated in Figure 1.

The chloroacetyl amides 2a–b were synthesised from commercially available amines 1a–b as previously described in the literature [27]. The reaction of intermediates 2a–b with thiazolidine-2,4-dione 3 provided the advanced intermediates 4a–b in acceptable yields [27]. Knoevenagel condensation of intermediates 4a–b with commercially accessible 2,4,6-trimethoxybenzaldehyde or 2,5-dimethyl-1-phenyl-1*H*-pyrrole-3-carbaldehyde [28] yielded the target thiazolidine-2,4-dione carboxamides 5a–b in good yields. Similarly, the target acid 7 was readily accessible via the conventional Knoevenagel condensation of the 2,5-dimethyl-1-phenyl-1*H*-pyrrole-3-carbaldehyde with the 3-thiazolidineacetic acid 6. Furthermore, the treatment of commercially available phenylthiourea 10 was achieved with chloroacetyl chloride under basic medium yield 2-(phenylimino) thiazolidine-4-one 9. Knoevenagel condensation of 9 with 2,5-dimethyl-1-phenyl-1*H*-pyrrole-3-carbaldehyde gave compound 10 (TZDD4) in good yield (Figure 2). The structural identity of each compound was confirmed using analytic spectroscopic methods (^1^H and ^13^C NMR, IR, and MS).

#### 4.1.1. Experimental Section

##### Materials and Physical Measurements

All commercially available chemicals and specialised reagents used in this study were purchased from Sigma-Aldrich^®^ (St. Louis, MO, USA) and Merck^®^ (Johannesburg, SA) and were used without further purification unless stated otherwise. The ^1^H and ^13^C NMR spectra were recorded in a designated solvent on a Bruker Fourier 300 MHz, Bruker Avance III HD 400 MHz, or Bruker Avance II 600 MHz spectrometer and were internally referenced to the solvent peaks (δ_H_ 7.6 and δ_C_ 77.0 ppm for CDCl_3_; δ_H_ 2.5 and δ_C_ 39.4 ppm for DMSO-*d_6_*). IR spectra were recorded on a Perkin-Elmer FT-IR Spectrum 100 spectrometer. High-resolution electrospray ionisation accurate mass measurements (HRMS-ESI) were recorded in positive or negative mode on a Waters Synapt G2 (Central Analytical Facility, University of Stellenbosch, South Africa). The melting points were determined on Reichert 281313 hot stage apparatus and are uncorrected. The progress of the reactions was monitored by analytical thin layer chromatography (TLC) using Merck F_254_ aluminium-backed pre-coated silica gel plates, which were visualized under ultraviolet (UV: 254 and 366 nm) light or, where necessary, stained in iodine vapour. The crude compounds were purified by silica gel column chromatography using Merck Kieselgel 60 Å (230–400 mesh ASTM) or by preparative thin-layer chromatography (PTLC) using Merck 60GF_254_ silica gel coated on glass plates (2.0 × 200 × 200 mm). Compounds **5b** (TZDD1) and **10** (TZDD4) were synthesised from their corresponding starting materials as previously described in the literature [11,27].

#### 4.1.2. General Synthetic Procedure Target Compounds **5a** (TZDD2) and **7** (TZDD3)

An equimolar mixture of 2,5-dimethyl-1-phenyl-1*H*-pyrrole-3-carbaldehyde and 3-(2-oxo-2-(pyrrolidin-1-yl)ethyl)thiazolidine-2,4-dione 4a or 3-thiazolidineacetic acid 6 in the presence of piperidine (1.5 eq) in ethanol (10 mL) was heated at 60 °C for 6 to 24 h. After the completion of the reaction (TLC), the solvent was removed in vacuo to give a crude reaction mixture, which was purified by silica gel column chromatography to afford the desired compounds (Figure 11) as solids.

##### (Z)-5-((2,5-Dimethyl-1phenyl-1H-pyrrol-3-yl)methylene)-3-(2-oxo-2-(pyrrolidin-1-yl)ethyl)thiazolidine-2,4-dione, 5a (TZDD2)

A yellow solid (57 mg, 79%); m.p. 256–258 °C; R_f_ (n-Hex: EtOAc; 50: 50%) 0.24; IR ν_max_/cm^−1^: 1729, 1661, 1584 (C=O), 1508 (Ar C=C); δ_H_ (300 MHz, CDCl_3_): 7.91 (1H, s, H_1_), 7.53–7.45 (3H, m, H_2,3_), 7.19 (2H, dd, *J* 8.0 and 1.7 Hz, H_4_), 6.22 (1H, s, H_5_), 4.44 (2H, s, H_6_), 3.51 (4H, td, *J* 6.8 and 4.3 Hz, H_7,7′_), 2.15 (3H, s, H_8_), 2.07–1.98 (5H, m, H_9,9′_), 1.88 (2H, quint, *J* 6.5 Hz, H_10_); δ_C_ (75 MHz, CDCl_3_): 169.0 166.7, 163.4, 137.6, 136.5, 132.1, 129.6 (2C), 128.9, 128.6, 128.0 (2C), 115.8, 113.4, 105.5, 46.3, 45.8, 43.0, 26.3, 24.2, 12.9, 11.3; HRMS: *m/z* (ESI) found 410.1532 [M + H]^+^, calcd. for C_22_H_24_N_3_O_3_S, 410.1538.

##### (Z)-2-(5-((2,5-Dimethyl-1-phenyl-1H-pyrrol-3-yl)methylene)-2,4-dioxothiazolidin-3-yl)acetic Acid, 7 (TZDD3)

A yellow solid (950 mg, 48%); m.p. 118–120 °C; R_f_ (DCM: 2M methanolic NH_3_; 90: 10%) 0.25; IR ν_max_/cm^−1^: 3381 (OH), 1720 (C=O, strained amide), 1655 (C=O, acid),1587 (C=O, thiolactone), 1498 (C=C); δ_H_ (400 MHz, CDCl_3_): 7.87 (1H, s, H_1_), 7.52–7.44 (3H, m, H_2,3_), 7.18 (2H, dd, *J* 8.4 and 2.0 Hz, H_4_), 6.20 (1H, s, H_5_), 4.28 (2H, s, H_6_), 2.14 (3H, s, H_7_), 2.02 (3H, s, H_8_); δ_C_ (100 MHz, CDCl_3_): 168.5, 167.3, 166.1, 137.5, 136.9, 132.3, 129.7 (2C), 129.0, 128.9, 128.0 (2C), 115.7, 112.7, 105.5, 44.3, 12.9, 11.2; HRMS: *m/z* (ESI) found 357.0913 [M + H]^+^, calcd. for C_18_H_17_N_2_O_4_S, 357.0909.

### 4.2. Biological Investigation

#### 4.2.1. α-Amylase and Glucosidase Inhibition Assay

The α-amylase inhibitory activity was determined using the method described by the Worthington Enzyme Manual with slight modifications [28,29]. The TZD derivatives were assayed at 10, 20, 30, 40, and 50 µg/mL. Acarbose (10, 20, 30, 40, and 50 µg/mL) was used as the positive control, and the absolute control contained all the reagents used except the inhibitor compounds. The sodium phosphate buffer was used as a blank. The inhibitory activity was calculated according to the equation:

Inhibition (%) = ((*A*_control_ − *A*_sample_)/*A*_control_) × 100%, where *A*_control_ was the absorbance of the control (without the inhibitor); *A*_sample_ was the absorbance in the presence of TZD derivatives.

#### 4.2.2. Determining Mode of Inhibition of Alpha Glucosidase

The mode of inhibition of the biological enzyme α-glucosidase, by the most potent derivative (TZDD3) in this study, was determined using the Michaelis-Menten and Lineweaver-Burk plots to determine the kinetic constants, as well as plots [30]. The inhibition of this enzyme activity was determined in the presence and absence of TZDD3 at a concentration of its IC_50_ and a concentration twice its IC_50_ (60 and 120 µg/mL, respectively). The Ki values were calculated with GraphPad Prism 9.2.0. (332) by plotting the reciprocal of maximum velocity (1/Vmax) (*y*-axis) against the derivative concentrations (*x*-axis). The types of inhibition parameters were all calculated with GraphPad Prism 9.2.0. Briefly, 30 µL of α-glucosidase enzyme (0.5 IU/mL) was dissolved in the 0.02 M phosphate buffer (pH 6.9), and then was pre-incubated at 37 °C with the above-mentioned derivative (50 µL) in a 96-well plate, for 5 min. In sequence, pNPG (0.125, 0.250, 0.500, 0.750, 1.000, 1.500, 2.000, 2.500, and 5.000 mM) was added and incubated in the reaction mixture at 37 °C for 30 min.

#### 4.2.3. Aldose Reductase Inhibition Assay

The aldose reductase (AR) inhibition assay was performed according to the method described by Kazeem et al. with minor modifications [31,32]. All the TZD derivatives were assayed at 10, 20, 30, 40, and 50 µg/mL. The positive control was performed using the same procedure but replacing the TZD derivatives with quercetin (10, 20, 30, 40, and 50 µg/mL). The absolute standard contained all the reagents used except for the inhibitor compounds. The aldose reductase inhibition activity was calculated as a percentage inhibition from:

% Inhibition = [(∆Abs_control_ − ∆Abs_sample_)/∆Abs_control_] × 100%, where ∆Abs is the change is absorbance.

#### 4.2.4. Protein Tyrosine Phosphatase Inhibition Assay

The protein tyrosine phosphatase-1B (PTP-1B) inhibition assay was performed according to the method described by Song et al. with some modifications [33,34]. Sodium orthovanadate (Na_3_VO_4_) (10, 20, 30, 40, and 50 µg/mL) was used as the positive control for inhibition. The absolute control contained all the reagents used except the inhibitor compounds. The inhibition activity was calculated using: 

% Inhibition = [(∆Abs_control_ − ∆Abs_sample_)/∆Abs_control_] × 100%, where ∆Abs is the change is absorbance.

#### 4.2.5. Dipeptidyl Peptidase-4 Inhibition Assay

This assay was performed according to the DPP-4 Inhibitor Screening Kit (catalogue number: MAK203) product information (Sigma-Aldrich.com) with minor modifications to the volumes to suit the 96-well plate assay.

#### 4.2.6. Glucose Uptake and Cell Viability

HEPG-2 liver cells were cultured in MEME supplemented with pen/strep (1%) and FBS (10%) in a 75 cm^3^ flask under tissue culture conditions. Upon confluency (80%), cells were trypsinized and seeded into the 96-well plates. Upon confluency, the cell preparations were exposed to different concentrations (25, 50, and 100 µg/mL) of TZDDs, and a media glucose reading was taken before incubation (time = 0). Thereafter, the plates containing cells and treatments were incubated for 24 h under tissue culture conditions. The non-treated cells and insulin treated cells served as absolute and positive controls, respectively. After a 24 h incubation, the medium glucose was taken using a glucometer [35]. The percentage glucose uptake was calculated as follows: 

Glucose uptake (%) = (T24/T0) × 100

To determine the cell viability, the acid phosphatase assay method was followed [36]. Briefly, after reading the media glucose concentration at 24 h, the old media was removed, and the cells were washed with phosphate buffered saline three times. In each well, 200 µL of a buffer (0.1 M sodium acetate, 0.1% Triton X-100, and 5 mM p-nitrophenol phosphate) was added. Afterwards, the plates were incubated at 37 °C for 2 h. Thereafter, the reaction was stopped by the addition of NaOH (1N, 10 µL), and the absorbance was read using a microplate reader. To eliminate the background colour, appropriate blanks were included. Cell viability was expressed as a percentage using the following formula:

Cell viability (%) = (Ab compound/Ab control) × 100

#### 4.2.7. Peroxisome Proliferator–Activated Receptor-γ Docking In Silico

The 3D crystal structure of peroxisome proliferator-activated receptor gamma (PPAR-γ) was downloaded from the Protein Data Bank (PDB) (https://www.rcsb.org/) [37]. The protein (PDB entry: 4Ema) for docking was prepared using the protein preparation wizard tools of AutoDock 4.2. The ligands (TZDD1, TZDD2, TZDD3, and TZDD4) were prepared using ChemDraw- AcdLabs software to generate the.dxt format and Openbabel-2.4.1 software to generate pbdqt formats. The grid points on the X, Y, and Z axis were set at 60 × 60 × 60. The grid centre was placed in the active site pocket centre, with the coordinates of Central Grid Point of Maps = −4.339, −14.270, 22.436). Minimum coordinates in grid = −19.339, −29.270, 7.436 and maximum coordinates in grid = 10.661, 0.730, 37.436.

The grid boxes included the entire binding site of this protein and provided enough space for the ligand’s translational and rotational walk. For each ligand, a docking experiment consisting of 50 stimulations was performed, and the analysis was based on binding free energies and root mean square deviation (RMSD) values. The ligand molecules were then ranked in order of increasing docking energies. Results were analysed using binding energy as well as ProteinPlus analysis tools software (https://proteins.plus/) and the Protein-Ligand Interaction Profiler (https://plip-tool.biotec.tu-dresden.de/plip-web/plip/index) to determine the binding conformations [38]. No further refinements of docking poses were conducted in the analysis.

### 4.3. Statistical Analysis

Each experiment was carried out in triplicates and repeated twice. Statistical analysis of results from each experiment was conducted using GraphPad Prism 9.2.0. Further statistical analysis was performed using a one-way ANOVA followed by a Tukey-Kramer post hoc test used to test the significance between a test compound and the absolute control. The statistical analysis was performed using absorbance or fluorescence readings before any normalization. The statistical significance was acknowledged at the *p* < 0.05 level. 

## 5. Conclusions

Taken together, our compounds show moderate activity in several targets for diabetes management. These observations may suggest that TZZDs can exert antidiabetic activity by inhibiting or activating various key targets in DM management, which could be of paramount therapeutic value. For these reasons, further investigations, including in vivo studies, are critical moving forward to gain holistic therapeutic insights.

## Data Availability

The data displayed in this study are available on request from the corresponding author.

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
