# Peer review of "Discovery of Novel Thiazolidinedione-Derivatives with Multi-Modal Antidiabetic Activities In Vitro and In Silico"

_ijms, 2023, doi:10.3390/ijms24033024_

Round 1
Reviewer 1 Report
Authors presented a study where they synthesized several TZD derivatives for the treatment of diabetes.
The results showed that these compounds are promising candidates for further studies and future alternatives to curren therapies.
The manuscript is well written, quite clear but I have some comments.
1. The compounds identification is not clear. In the results figures compounds are listed as TZDD2 etc however in the experimental section this nomenclature is not observed and it is not clear which compound correspond to those in the graphs. This must be cleared by the authors.
2. In Line 43: Although some type2 diabetes are adminstrered in type1 patients, all these drugs are tipical of type 2 diabetes. Therefore, this should be clarified. For some that is not familiar with diabetes treatments may think that this compounds could be used for T1D.
3. In line 46: Type 2 Diabetes should be written instead of only Diabetes, because there are different types and the treatments are different.
4. In line 50: "." must be replaced by ","
5. In line Figure 9, 10. axes labels are too thick.
After this minor revisions I believe that the manuscript is ready for publication as it gives a new insight on T2D therapies.
Author Response
Please see attachment in the box

Reviewer 2 Report
The discovery and characterization of compounds showing antidiabetic activities is a highly relevant endeavor in the current times with rapidly escalating numbers of individuals affected by the serious medical condition of diabetes. Seen in that context, the present investigation appears as being timely as it adds to our understanding of the design of suitable pharmaceutical treatment modalities. Unfortunately, however, the compounds prepared, analogues of a well known structural class, and thoroughly investigated display only moderate activity and do, therefore, not offer any breakthrough in terms of therapeutic opportunities. This clearly reduces the value of this work, but publishing the results achieved can still be motivated as it adds to the understanding of the action and activity of the thiazolidindione series of compounds.
Before publishing this paper, several improvements have to be undertaken. For example, as noted in the attachment in the form of the amended manuscript, there are straighout errors to be corrected (the authors are strongly recommended to conduct a much more diligent proof-reading to avoid the many unnecessary typos before submission!). Furthermore the readability of several of the illustrations has to be addressed (again something the authors should have done right from the start and not awaiting comments/criticism from a referee!). In the list of literature references, the cited journal names are given with their full title, which stands in stark contrast to the the normal abbreviated format recommended by most (all) scientific journals. Following these principles is a strong recommendation here as well.

Author Response
Please see the attachment in the box

Reviewer 3 Report
The manuscript "DISCOVERY OF NOVEL THIAZOLIDINEDIONE-DERIVATIVES WITH MULTI-MODAL ANTIDIABETIC ACTIVITIES IN VITRO AND IN SILIC," by Arineitwe et al describes the synthesis, testing and computational analysis of several compound against proteins involved in diabetes. It is interesting and important work but needs improvement in the computational portion. I will comment mainly on the computational portion as that is my speciality, but I will say a bit about the experimental portion as well. As it is currently written I cannot recommend publication.
The protein assays conducted with the novel compounds seem to have been done and presented well, but little comparative analysis is presented. The authors should present a table or text describing the selectivity of the various compounds. Also, the text should include a figure with chemdraw (or similar) structures for the reference compound and each novel compound.
Below I will list areas that should be improved in the computational portion:
1. The conformations used seem to have been taken from docking poses with no further refinement. These structures should be optimized further.
2. In the figure of the conformations, all of the active sites should be in the same orientation so that the reader can see the differences in conformations. As is, the figure is almost impossible to interpret.
3. An interaction is described as "hydrophobic" though this is not a type of interaction in the same sense as "hydrogen bonds" or pi-stacking." The authors should be more careful with this description.
4. There is a residue that is alternately labelled at Leu128 and Leu 228. Is this the same residue or 2 different residues?
5. The table described hydrogen bonds between the novel compounds and Leu and Ile residues, though the images do not show hydrogen bonds, In fact, unless binding is occurring to the protein backbone then these residues would not form hydrogen bonds. Further, it is made more difficult to interpret the interactions since the colour in in the conformations does not correspond to atom types. In fact, no legend is provided to say why the atoms are coloured as they are and so makes the figure even more difficult to interpret.
6. Finally, a table should be made to show how well the calculated values correspond to experimental values.
Author Response
Please see the attachment in the box

Round 2
Reviewer 2 Report
The revision has been conducted in a satisfactory manner by making the corrections demanded by referees. This has alltogether brought the paper to a level of quality where it meets reasonable criteria. Thus, thie work is now ready for being accepted and published.
Reviewer 3 Report
The revised manuscript "DISCOVERY OF NOVEL THIAZOLIDINEDIONE-DERIVATIVES WITH MULTI-MODAL ANTIDIABETIC ACTIVITIES IN VITRO AND IN SILIC," by Arineitwe et al has been greatly improved. The previous concerns I had about the clarity of the results and interpretation of forces have been addressed satisfactorily. The only remaining concern I have is that docking poses were not further refined in the analysis, but if the authors clearly state this, then I can now recommend the manuscript for publication.
